# Ceftriaxone for the Treatment of Chronic Bacterial Prostatitis: A Case Series and Literature Review

**DOI:** 10.3390/antibiotics11010083

**Published:** 2022-01-11

**Authors:** Sander G. Kuiper, Maarten Ploeger, Erik B. Wilms, Marleen M. van Dijk, Emiel Leegwater, Robert A. G. Huis in’t Veld, Cees van Nieuwkoop

**Affiliations:** 1Department of Internal Medicine, Haga Teaching Hospital, 2545 AA The Hague, The Netherlands; c.vannieuwkoop@hagaziekenhuis.nl; 2Department of Hospital Pharmacy, Haga Teaching Hospital, 2545 AA The Hague, The Netherlands; m.ploeger@hagaziekenhuis.nl (M.P.); e.wilms@ahz.nl (E.B.W.); E.Leegwater@hagaziekenhuis.nl (E.L.); 3Department of Urology, Haga Teaching Hospital, 2545 AA The Hague, The Netherlands; m.m.vandijk@hagaziekenhuis.nl; 4Department of Medical Microbiology, University of Groningen, University Medical Center Groningen, 9713 GZ Groningen, The Netherlands; r.a.g.huis.in.t.veld@umcg.nl

**Keywords:** ceftriaxone, chronic bacterial prostatitis, *Escherichia coli*

## Abstract

Chronic bacterial prostatitis is increasingly difficult to treat due to rising antimicrobial resistance limiting oral treatment options. In this case series, 11 men with CBP (including patients with urological comorbidities) due to multi-resistant *E. coli* were treated with once-daily ceftriaxone intravenously for 6 weeks. Nine patients were clinically cured at 3 months follow up. No early withdrawal of medication due to side effects occurred. A literature review was conducted to describe the prostate pharmacokinetics of ceftriaxone and its use in prostatic infection. In conclusion, ceftriaxone can be considered an appropriate treatment of chronic bacterial prostatitis.

## 1. Introduction

Chronic bacterial prostatitis (CBP) is characterized by recurrent urinary tract infection (UTI) caused by the same uropathogen originating from the prostate [1,2,3]. Typical symptoms include urinary frequency, urgency, dysuria and perineal pain; however, it may also be accompanied with fever, malaise, and occasionally the urosepsis syndrome [1,4]. The ultimate goal of antibiotic therapy is to eradicate the causative uropathogen; however, as the inflammation resolves, the antibiotic concentrations that can be reached in prostatic tissue decreases, which limits successfully reaching this goal [5]. To penetrate the prostate sufficiently, the chosen antibiotic must ideally have a high lipid solubility, low degree of ionization, low protein binding and a high dissociation constant [4,5,6]. Because of their excellent pharmacokinetic and pharmacodynamic characteristics, fluoroquinolones for 4–6 weeks are therefore the preferred treatment for CBP with clinical and microbiological cure rates of 60–80% [1,4,7,8]. Treatment with trimethoprim-sulfamethoxazole is considered a good oral alternative, although less effective than fluoroquinolones [1,4]. Uropathogen resistance to fluoroquinolones and trimethoprim-sulfamethoxazole is increasing, limiting treatment options of CBP. The available evidence supporting treatment with other antibiotics comprises small case series or cohort studies [1]. Usually, individual antibacterial treatment schedules will be selected based on the resistance pattern of the causative uropathogen and specific patient characteristics such as renal function and possible allergies [3]. Beta-lactam antibiotics (administered either orally or intravenously) are generally considered of little value for the treatment of CBP, as most of them achieve poor prostatic tissue levels. The 3rd generation cephalosporin ceftriaxone is an exception showing excellent and prolonged prostatic tissue concentration after a single dose of 2 g intravenously [5,9]. As such, intravenous ceftriaxone once daily as outpatient parenteral antimicrobial therapy (OPAT) is considered a potential treatment option for CBP when fluoroquinolones and trimethoprim-sulfamethoxazole therapy are contra-indicated (e.g., due to resistant causal uropathogen and/or allergy) [4,10]. To our knowledge, to date there are no studies reporting on the efficacy and safety of ceftriaxone in CBP. Based on our OPAT experience in recent years, we performed a retrospective analysis of CBP patients treated with ceftriaxone with the aim to assess the clinical and microbiological outcome. In addition, we reviewed the literature upon the pharmacokinetics of ceftriaxone in the prostate and its efficacy as treatment of CBP.

## 2. Results

Thirteen patients were treated with ceftriaxone in OPAT setting for presumed CBP, two of them did not meet the inclusion criteria. One patient did not meet the definition of CBP and one patient did not have primary end point data. Eleven patients were included in the final analysis. The demographics are shown in Table 1. Mean age (range) was 72 (52–86) years. Seven out of eleven patients had a urologic history, mostly prostate carcinoma. The causative micro-organism was *E. coli* in all patients and all positive urinary cultures had a growth of >10^5^ colony forming units (CFU) per mL urine. All *E. coli* isolates were resistant to fluoroquinolones and trimethoprim-sulfamethoxazole, none were resistant to fosfomycin and none produced extended beta-lactamase (ESBL). Three out of eleven patients were treated with fosfomycin before they started with ceftriaxone treatment; all of them had recurrent UTI with the same *E. coli* after discontinuation of fosfomycin. The mean Charlson comorbidity index value was 5 (range 1–9). Mean duration of ceftriaxone treatment was 40 days (range 28–45). Nine out of eleven (82%) patients had clinical cure at 3 months. The microbiological cure rate was 8 out of 9 and 7 out of 9 at 1 and 3 months, respectively. The late clinical cure rate was 7 out of 8 at 6 months; 1 clinical and microbiological failure occurred between 3 and 6 months. None of the patients prematurely discontinued ceftriaxone treatment due to adverse events. No phlebitis of the peripheral intravenous central catheter (PICC) occurred; one patient had injury of the ulnar nerve related to placement of the PICC catheter. 

## 3. Literature Review

After reviewing 59 studies, 8 met our inclusion criteria; four studies investigated the pharmacokinetics of ceftriaxone in the prostate; four studies were found reporting findings about ceftriaxone treatment to prevent urinary tract infection and/or prostatitis after prostatic biopsy or surgery. 

The first pharmacokinetic study reported ceftriaxone concentrations in prostatic fluid. Twelve patients received a single dose of ceftriaxone (1 g, intramuscular); prostatic fluid ceftriaxone concentration (measured by bioassay) was between 1.17 (±0.58) ug/mL after 3 h and 0.23 (±0.11) ug/mL after 6 h [11]. In contrast with the first study, the other pharmacokinetics studies measured ceftriaxone concentration in prostatic tissue after taking a biopsy specimen during prostatic surgery. In 46 patients (aged 59–84 years), prostate adenoma tissue concentrations (measured by bioassays) were 12.9–73.7 ug/g if ceftriaxone (2 g, intravenously at different time points before prostatectomy) was administered 30 min before the biopsy and 0.5–19.3 ug/g 24 h after administration. The tissue half-life was 8 h. Though there was a wide inter-individual and intra-individual (differences between concentration in left and right prostate lobe) variety in prostate levels, the authors concluded that ceftriaxone reaches sufficient concentrations to treat infections of the prostate [9]. Another study included seven men while being treated with ceftriaxone (1 g twice daily, intramuscular). The prostatic ceftriaxone concentrations (measured by high-performance liquid chromatography) ranged from 6.95 ug/g to 16.52 ug/g. The mean ceftriaxone prostate/serum ratio was 0.37 (SD ± 0.12). In this study, it was unclear how long after the last dose of ceftriaxone, prostate samples were taken [12]. During open prostatectomy in 15 patients (aged 67 ± 7 years), different tissue samples were taken, including the prostate. Approximately 64 min (SD ± 10) after administration of ceftriaxone (1 g, intravenously 30 min before prostatectomy), the mean prostate tissue concentration (measured by high-performance liquid chromatography) was 35 (SD ± 18) ug/g and the mean prostate/serum concentration was 0.38 (SD ± 0.18), which is well above the MICs of common uropathogens. In this study, before determination of tissue concentration, 100 mg of prostatic tissue was crushed with 1 milliliter of isotonic saline solution. This could have led to a wide spread of the tissue concentration [13]. Currently, micro-dialysis is used to counter this variation in determined tissue concentration; the reported values could be seen as estimates of the real value [14].

There are no studies reporting the clinical or microbiological efficacy of ceftriaxone in CBP. The only available studies are studies on prophylactic use of ceftriaxone prior to transrectal prostate biopsy to prevent infection. The first report on ceftriaxone use in prophylactic treatment after prostate biopsy is an open, randomized trial in which 101 patients were randomized to no prophylaxis versus two dosages of ceftriaxone (1 g, intravenously) pre-biopsy. This resulted in a reduction of 14 to 3 patients with bacteriuria and symptoms requiring antibiotics [15]. In 5577 patients, adding ceftriaxone (2 g, intravenously) on top of ciprofloxacin as prophylaxis before prostate biopsy reduced the amount of procedure related infections significantly from 2.31% to 0.2% [16]. This effect was also shown in 4143 performed prostate biopsies. One dose of ceftriaxone (1 g, intravenously) and one dose of 500 mg ciprofloxacin pre-biopsy compared to ciprofloxacin for 4 days (started 1 day pre-biopsy, 500 mg twice daily) resulted in a reduction of the post biopsy hospitalizations from 14 (0.6%) to 0 (0%) in favor of the ceftriaxone group [17]. In a prospective cohort study, ceftriaxone combined with ciprofloxacin was compared with ciprofloxacin and gentamicin as pre-biopsy prophylaxis in 829 patients. This resulted in a reduction of the incidence of post-biopsy sepsis from 12 (3.8%) in the ciprofloxacin/gentamycin group to 4 (2%) in the ceftriaxone/ciprofloxacin group [18].

## 4. Discussion

In this study, we showed that ceftriaxone is a viable treatment option for *E. coli* chronic bacterial prostatitis when fluoroquinolones and trimethoprim-sulfamethoxazole cannot be used. The achieved clinical cure rate of 80% is comparable to treatment with fluoroquinolones or trimethoprim-sulfamethoxazole, of which cure rates at 3–6 months after treatment range from 60–80% [7]. To date, to our knowledge, this is the first study to show clinical efficacy of ceftriaxone in patients with CBP. Ceftriaxone is one of the most frequently used antibiotics in outpatient parenteral treatment because of convenient once daily dosing, favorable side effects and drug stability after reconstitution. None of the patients had to discontinue ceftriaxone treatment due to adverse events and no phlebitis occurred. These results confirm feasibility and clinical efficacy of outpatient ceftriaxone treatment of patients with CBP.

In our clinical practice located in The Netherlands, combined fluoroquinolones and TS resistance in gram-negative uropathogens is still relatively rare, which is represented in the small number of patients requiring outpatient ceftriaxone treatment in the past eight years. However, increasing rates of fluoroquinolones and trimethoprim-sulfamethoxazole resistance is reported worldwide [19,20,21,22]. Assuming this also affects patients with CBP, this urges the need to find alternative treatment options. Another alternative treatment option which has recently received increasing attention as treatment for CBP is fosfomycin [23,24]. In our patient cohort, all isolates were susceptible to fosfomycin. Some patients in our cohort were treated with oral fosfomycin but without resolution of CBP; they subsequently were successfully treated with ceftriaxone. Further studies are needed to elucidate the optimal effective dose, its pharmacokinetics, and oral tolerability in daily dosing to assess the clinical usefulness of oral fosfomycin in the treatment of CBP.

We used resolution of symptoms and absence of UTI recurrence during 3-month follow-up as the primary endpoint. To date, there is no validated test of cure for bacterial prostatitis [4]. There was one patient who had a clinical failure after our primary end point of 3 months. Whether this is caused by re-infection, sustained risk factors or persistence of the infection between 3 and 6 months cannot be concluded from this data. Therefore, when conducting studies on the treatment of CBP, we recommend using extended follow-up after treatment of CBP as clinical endpoint, to be able to address the issue of late clinical failures. Moreover, lower urinary tract cultures such as the Meares Stamey four-glass or the two-glass test to confirm the diagnosis of CBP were not used in this population, because this is a selected group of patients who already have had repeated positive urinary cultures with the same uropathogen. Reported specificity of positive urinary culture for having CBP is 100% [25]. Our clinicians considered the patients as having CBP and adjusted the treatment duration to the according guidelines being 6 weeks.

The strengths of this cohort study are its straightforward design reflecting daily practice of treating CBP in elderly men with significant underlying urologic comorbidities. However, its limitations are the small number of patients and its retrospective design. Therefore, the reported efficacy of ceftriaxone is only an indication of the true effect which needs to be further elucidated. Furthermore, it should be emphasized that a diagnosis of CBP was not confirmed by lower urinary tract cultures. As such, a definite distinction between CBP or complicated (recurrent) UTI could not be made in the described cases. Indeed, all cases fulfilled the criteria for CBP, complicated UTI and recurrent UTI as defined by the guideline of the European Association of Urology [3]. 

Our literature review revealed several pharmacokinetic studies that all showed therapeutic concentrations of ceftriaxone into the prostate. Furthermore, ceftriaxone has been shown to be clinically effective in the prevention of urinary tract infection, including prostatitis, when used as prophylaxis prior to prostate biopsy. All together, these data support our conclusion that ceftriaxone is an appropriate treatment for patients with chronic bacterial prostatitis caused by susceptible uropathogens. 

## 5. Materials and Methods

We conducted a single center study in the outpatient clinic of the Haga Teaching Hospital in The Hague, The Netherlands. Ethical approval exempting applicability of the Medical Research Involving Human Subjects Act to this study was received (number 19–020). The study was approved by the Institutional Scientific Review Board (number 2019/93/JW). From 2012–2020 all patients with CBP with a microorganism resistant to fluoroquinolones and trimethoprim-sulfamethoxazole but susceptible to ceftriaxone, were treated as per local protocol with 6 weeks of ceftriaxone 2 g intravenously every 24 h via a PICC. Patients were selected from a database of the OPAT program. OPAT was used in above mentioned protocol for the intravenous treatment at home [26].

Inclusion criteria were men aged >18 years old with a diagnosis of CBP and treatment with ceftriaxone for 6 weeks. Diagnosis of CBP was defined as recurrent UTI with the same uropathogen without any other abnormalities within the urinary tract that might explain recurrent UTI. Midstream urinary cultures were used, and positive urinary culture was defined as isolation of an uropathogen with a bacterial growth over 10^3^ CFU per mL urine [27,28]. The sole exclusion criterion was absence of follow up at the primary end point.

The primary end point was clinical cure rate of CBP at 3 months, defined as absence of recurrent UTI (no additional antibiotic treatment) and resolution of UTI symptoms (e.g., urinary frequency, urgency, dysuria or perineal pain). Secondary endpoints were clinical cure rate at 6 months, microbiological cure at 1 and 3 months and adverse events leading to discontinuation of therapy. Microbiological failure was defined as a positive urine culture with a similar uropathogen as found before ceftriaxone treatment. Similarity was determined based on the species as identified by mass spectrometry, growth characteristics and phenotypic susceptibility pattern. 

For the literature review, the database of PubMed was searched with the keywords prostate and ceftriaxone. The search strategy was: (“Prostate”(Mesh) OR “Prostatitis”(tw) OR “Prostate”(majr) OR “Prostate”(ti) OR “Prostates”(ti) OR “prostatic”(ti)) AND (“ceftriaxone” (tiab) OR “ceftriaxone” (MeSH Terms)) (accessed on 30 November 2021).

The search strategy resulted in 59 articles, which were screened for eligibility based on the following inclusion criteria: 1. Studies investigating ceftriaxone concentration in human prostate tissue or prostate fluid 2. Studies reporting the clinical and microbiological effect of ceftriaxone in prostate infections. We selected all articles in English and excluded case reports and articles regarding treatment for gonococcal urinary tract infections.

## 6. Conclusions

In conclusion, this study shows that ceftriaxone (2 g once daily, intravenously), preferably in an outpatient setting, could be considered an appropriate treatment for patients with chronic bacterial prostatitis. Future studies are warranted to definitively assess its clinical effectiveness.

## Figures and Tables

**Table 1 antibiotics-11-00083-t001:** Characteristics of patients with CBP treated with ceftriaxone 2 g every 24 h for 6 weeks.

	Age (Years)	Urologic Comorbidity	Pretreatment with Oral Fosfomycin	Days of OPAT	Outcome at 3 Months	Outcome at 6 Months
1	76	Nephrectomy, cured bladder carcinoma by BCG treatment, renal transplant	None	42	Clinical/microbiological cure	Clinical/microbiological failure
2	82	Cured prostatic carcinoma	None	35	Clinical/microbiological failure	NA
3	82	Cured prostatic carcinoma, placing gold markers	Yes, 14 days	42	Clinical/microbiological cure	Clinical cure, microbiological NA
4	72	None	None	33	Clinical cure, microbiological NA	Clinical cure, microbiological NA
5	72	Metastatic prostatic carcinoma	None	43	Clinical/microbiological cure	Clinical cure, microbiological NA
6	56	BPH, prostate biopsy	None	42	Clinical/microbiological cure	Clinical cure, microbiological NA
7	86	BPH, TURP, urolithiasis	None	43	Clinical/microbiological failure	NA
8	52	CIC because of areflexia bladder	None	45	Clinical/microbiological cure	Clinical cure, microbiological failure
9	57	None	None	42	Clinical/microbiological cure	NA
10	70	None	6 weeks fosfomycine; 1 week 3 g every 24 h, 5 weeks 3 g every 48 h	28	Clinical/microbiological cure	Clinical cure, microbiological NA
11	67	None	3 g every 72 h for 9 months	42	Clinical cure/microbiological NA	Clinical cure, microbiological NA

OPAT: outpatient parenteral antimicrobial treatment; BPH: benign prostatic hyperplasia; BCG: Bacille Calmette-Guerin; TURP: transurethral resection of the prostate; CIC: clean intermittent catherization; NA: not available; h: hours.

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
