# Peer review of "Ceftriaxone for the Treatment of Chronic Bacterial Prostatitis: A Case Series and Literature Review"

_antibiotics, 2022, doi:10.3390/antibiotics11010083_

Round 1

Reviewer 1 Report

Dear Authors,

The manuscript submitted by Sander G. Kuiper et al., entitled "Ceftriaxone for treating chronic bacterial prostatitis: a case series and literature review," is quite interesting. CBP treatment is a challenge for every urologist, mainly due to the long treatment period and poor results because of low penetrating the prostate. 

Here are my comments:

  1. l. 15 review of the ceftriaxone in prostate infections??
  2. l. 21-22 the definition of CBP is not adequate.
  3. Please put the reference before the point [x].
  4. l. 22-27 needs a reference
  5. l. 37-39 needs a reference
  6. l. 55 3 patients had prostate carcinoma and CBP? Maybe this patients had recurrent cUTI (complicated urinary tract infection)?
  7. l. 150-151 Please explain the diagnosis of CBP is based on what, according to which guidelines?
  8. l. 167 Please provide the number
  9. Material and methods - The diagnosis used for CBP is not adequate. CBP - Chronic or recurrent urogenital symptoms with evidence of bacterial infection of the prostate.
    Krieger JN, Nyberg L Jr, Nickel JC. NIH consensus definition and classification of prostatitis. JAMA. 1999;282(3):236-237. doi:10.1001/jama.282.3.236
    "Bacterial prostatitis is a clinical condition caused by bacterial pathogens. It is recommended that urologists
    use the classification suggested by the National Institute of Diabetes, Digestive and Kidney Diseases (NIDDK)
    of the National Institutes of Health (NIH), in which bacterial prostatitis, with confirmed or suspected infection, is
    distinguished from chronic pelvic pain syndrome (CPPS)".
    EAU Guidelines. Edn. presented at the EAU Annual Congress Milan Italy 2021. ISBN 978-94-92671-13-4.
  10. l. 177 the standard for positive urine culture is a bacterial growth over 100,000 CFU. Please explain the rationale used for over 1,000 CFU.
  11. l. 179, please explain how the authors evaluate the clinical cure rate. They have not used any prostatitis symptom questionnaire, like Chronic Prostatitis Symptom Index (CPSI). The CPSI is validated for CBP.
  12. l. 182-183 please define clearly the positive urine culture.

Author Response

1. 15 review of the ceftriaxone in prostate infections??

Authors: We describe the pharmacokinetics of ceftriaxone in the prostate and the use of ceftriaxone in prostatic infections. For more clarity we adjusted the sentence to: “A literature review was conducted to describe the pharmacokinetics of ceftriaxone and its use in prostatic infections.”

2. 21-22 the definition of CBP is not adequate.

Authors: The sentence is adjusted to “Chronic bacterial prostatitis (CBP) is characterized by recurrent urinary tract infection (UTI) caused by the same uropathogen originating from the prostate.”

3. Please put the reference before the point [x].

Authors: This is corrected throughout the manuscript.

4. 22-27 needs a reference

5. 37-39 needs a reference

Authors: We have added references to these sentences.

6. 55 3 patients had prostate carcinoma and CBP? Maybe this patients had recurrent cUTI (complicated urinary tract infection)?

Authors: We assume that the question implies that the patients got a prostatectomy. 2 patients had cured prostatic cancer, after treatment with external brachy radiotherapy (EBRT). The patient with metastatic prostatic carcinoma haven’t had a prostatectomy. By defition (see inclusion criteria), all described patients had recurrent UTI with the same uropathogen (E. coli) that was judged (by the treating urologist and infectious disease specialist) to have been originated from the prostate as there were no other underlying urologic disorders (e.g. renal stones) to explain the recurrent UTI.         

7. 150-151 Please explain the diagnosis of CBP is based on what, according to which guidelines?

Authors: See our comment on question 9 below.

8. 167 Please provide the number

Authors: The numbers of the medical ethical committee waiver and the approval of the institutional scientific board are added.

9. Material and methods - The diagnosis used for CBP is not adequate. CBP - Chronic or recurrent urogenital symptoms with evidence of bacterial infection of the prostate.
Krieger JN, Nyberg L Jr, Nickel JC. NIH consensus definition and classification of prostatitis. JAMA. 1999;282(3):236-237. doi:10.1001/jama.282.3.236
"Bacterial prostatitis is a clinical condition caused by bacterial pathogens. It is recommended that urologists
use the classification suggested by the National Institute of Diabetes, Digestive and Kidney Diseases (NIDDK)
of the National Institutes of Health (NIH), in which bacterial prostatitis, with confirmed or suspected infection, is
distinguished from chronic pelvic pain syndrome (CPPS)".
EAU Guidelines. Edn. presented at the EAU Annual Congress Milan Italy 2021. ISBN 978-94-92671-13-4.

Authors: Question 7 and 9 are questions about the use of the definition of chronic bacterial prostatitis. The distinction between chronic prostatic pain syndrome and CBP is relevant in patients with complaints of the urinary tract, and without a positive urinary culture. A four-glass test (or two-glass test) is necessary to distinguish between both entities. In our population, all patients had positive urinary cultures, and we concluded that they had chronic bacterial prostatitis as there was no other urologic abnormality to explain the recurrent UTI. The EAU guideline which is referenced advises to perform a 4- or 2-glass test to categorize clinical prostatitis, but does not provide an advice in our specific population with already positive urinary culture. There could be a misclassification in this approach. In the discussion we appraised this topic as well in line 146-151.

10. 177 the standard for positive urine culture is a bacterial growth over 100,000 CFU. Please explain the rationale used for over 1,000 CFU.

Authors: If the bacterial growth is over 100.000 a definite urinary tract infection can be diagnosed. An urinary culture with a 1000 CFU could also be defined as positive when symptoms of urinary tract infections exist. As we use a urinary culture as one of the outcomes, we decided to use the lower bacterial growth to define positive urinary culture to not miss any urinary tract infections. At baseline, all of the included patients had a urinary culture with a growth of 100.000 CFU. We added this sentence to the manuscript to be clear, at line 59-60.

11. 179, please explain how the authors evaluate the clinical cure rate. They have not used any prostatitis symptom questionnaire, like Chronic Prostatitis Symptom Index (CPSI). The CPSI is validated for CBP.

Authors: This study has a retrospective design, and the CPSI is not used in the daily practice in our hospital. So it was not possible to use the questionnaire.

Thanks for the comment, we agree that the method section was not clear enough. We adjusted this section, also on question 12 and the comment of reviewer 2. The clinical cure rate was evaluated using the patients chart. In case of absence of any complaints of urinary tract infections, we reported a patient as clinical cured.  

12. 182-183 please define clearly the positive urine culture.

Authors: We stated in line 175-177 that a positive urinary culture is defined as a bacterial growth over 103 colony forming units per mL urine. In addition to this, the isolated pathogen must be similar to the pathogen cultured before treatment with ceftriaxone.

We have added references (Rubin et al and Wilson et al) of literature that discuss to use 103 colony forming units per mL urine as a cut off for significant bacterial growth in men

Reviewer 2 Report

Good and well written paper. It is of good quality and especially interesting from a clinical perspective. The combination of case reports and literature review could be of value for clinicians treating these patients in countires with low rate of ESBL.

Major concerns:

  • Methodology section could benefit from the CARE checklist.

The CARE Guidelines: Consensus-based Clinical Case Reporting Guideline Development | The EQUATOR Network (equator-network.org)

  • "Informed Consent Statement: Not applicable."

The authors state that they followed their clinical practice guideline” in our clinical practice located in The Netherlands”. However, is not informed consent needed in clinical studies in the Netherlands, even if only in observational, case report settings? At least this should be clearly stated and why this is not needed.

Minor concerns:

  • Some language edits could suffice.
  • Although stated in the main text. The manuscript, including abstract and title, could in general include clearer information on that these 11 cases described were of men with high co-morbidity including urological co-morbidities.

Author Response

Good and well written paper. It is of good quality and especially interesting from a clinical perspective. The combination of case reports and literature review could be of value for clinicians treating these patients in countries with low rate of ESBL.

Major concerns:

  • Methodology section could benefit from the CARE checklist.

The CARE Guidelines: Consensus-based Clinical Case Reporting Guideline Development | The EQUATOR Network (equator-network.org)

Authors: We agree that the method section was not clear enough. We adjusted the methods to make it more clear. It was not possible to use the CARE guideline, as it is made for case reports, and would extensively lengthen the manuscript without adding relevant content.

  • "Informed Consent Statement: Not applicable."

The authors state that they followed their clinical practice guideline” in our clinical practice located in The Netherlands”. However, is not informed consent needed in clinical studies in the Netherlands, even if only in observational, case report settings? At least this should be clearly stated and why this is not needed.

Authors: As stated in the Dutch Law, in the Medical Research Involving Human Subjects Act (WMO) it is stated that an informed consent is needed if it involves human subjects and if the subjects are subjected to actions or behavioral or rules apply on them. This was not the case in our study as the subjects were already treated by their physician and not by means of the study; it was retrospective. We confirmed this by getting a waiver for applicability of the WMO from the Medical Ethical Committee.

Minor concerns:

  • Some language edits could suffice.
  • Although stated in the main text. The manuscript, including abstract and title, could in general include clearer information on that these 11 cases described were of men with high co-morbidity including urological co-morbidities.

Authors: Thanks for this suggestion. We changed the abstract to make it more clearer. In addition, our manuscript was reviewed by a native speaker for language edits. 

Reviewer 3 Report

This is a well-written paper pertaining to an important topic. There are no obvious shortcomings and this paper can be accepted as is.

Author Response

Authors: Thanks for the approving review, we hope the reviewer is also satisfied with the adjustments made in correction to the comments of the other reviewers.

Round 2

Reviewer 1 Report

Dear Authors,

First of all, I want to congratulate you on your work. The manuscript is improved, but some aspects regarding CBP diagnosis still need to be addressed by the authors. 

CBP - Chronic or recurrent urogenital symptoms with evidence of bacterial infection of the prostate. There is no evidence of bacterial infection of the prostate in the study group, and there are only suppositions. According to the available guidelines, all the patients presented recurrent cUTI (complicated urinary tract infection).

Please clearly explain these aspects in the manuscript and include these in the limitations of the study. 

Author Response

First of all, I want to congratulate you on your work. The manuscript is improved, but some aspects regarding CBP diagnosis still need to be addressed by the authors. 

CBP - Chronic or recurrent urogenital symptoms with evidence of bacterial infection of the prostate. There is no evidence of bacterial infection of the prostate in the study group, and there are only suppositions. According to the available guidelines, all the patients presented recurrent cUTI (complicated urinary tract infection).

Please clearly explain these aspects in the manuscript and include these in the limitations of the study. 

Authors: Indeed, the definitions of UTI as outlined by international guidelines (e.g. European Association of Urology, EAU) might be confusing.

With respect to prostatitis, the EAU guideline refers to the NIH Consensus Definition and Classification of Prostatitis (Krieger JN et al, JAMA; 1999: 236-237). In this classification, Chronic Bacterial Prostatitis (CBP) is characterized by ‘recurrent episodes of bacterial urinary tract infection caused by the same organism, usually E. coli, another Gram-negative organism, or enterococcus. Between symptomatic episodes of bacteriuria, lower urinary tract cultures can be used to document an infected prostate gland as the focus of these recurrent infections’.

In our study, we included men with recurrent UTI due to the same E. coli while there was no other reasonable focus (as judged by the treating urologist and/or infectious diseases specialist) than CBP to explain the recurrent UTI.

As our inclusion criteria fulfill the above-named definition of CBP, we disagree with this reviewer that the described cases in our study did not suffer from CBP. Indeed, we did not perform so-called lower urinary tract cultures (e.g. the Meares Stamey four-glass or the two-glass test) to microbiologically confirm the prostatic gland was infected. Thus, we understand this reviewer might have some doubts upon a diagnosis of CBP. Though this has already been described in the discussion section of our manuscript, we have now highlighted this as a potential limitation. Nevertheless, it should be emphasized that guidelines consider lower urinary tract cultures to be supportive but not essential to make a diagnosis of CBP.

The EAU guideline on urogenital infections defines complicated UTIs as ‘all UTIs which are not defined as uncomplicated. Meaning in a narrower sense UTIs in a patient with an increased chance of a complicated course: i.e. all men, pregnant women, patients with relevant anatomical or functional abnormalities of the urinary tract, indwelling urinary catheters, renal diseases, and/or with other concomitant immunocompromising diseases for example, diabetes.’

In this respect, by definition all men with CBP should be considered to have complicated UTI.

Furthermore, recurrent UTI is defined as ‘recurrences of uncomplicated and/or complicated UTIs, with a frequency of at least three UTIs/year or two UTIs in the last six months’. As such, the included men in our study did also meet the criteria for recurrent UTI.

So, based on these definitions we might conclude that the described cases in our study all had complicated recurrent UTI due to E. coli. However, we consider CBP is the best term to be used as this is most likely clinical diagnosis and thus clinicians would treat such patients with antibiotics that penetrate the prostate sufficiently.